# Low Dose SPECT Image Denoising Using a Generative Adversarial Network

**Qi Zhang**[1]                                                                                          MB85402@UM.EDU.MO
[1] *Department of Computer and Information Science, Faculty of Science and Technology, University of Macau, Macau SAR, Peoples Republic of China*
**Jingzhang Sun**[2]                                                                                          YB87437@UM.EDU.MO
[2] *Department of Electrical and Computer Engineering, Faculty of Science and Technology, University of Macau, Macau SAR, Peoples Republic of China*
**Greta S. P. Mok**[2,3]                                                                                          GRETAMOK@UM.EDU.MO
[3] *Faculty of Health Sciences, University of Macau, Macau SAR, Peoples Republic of China*

## 1. Introduction

Single-photon emission computed tomography (SPECT) is an in vivo functional imaging technique that uses gamma cameras to detect molecular-level activities of patients tissues generally through injection of the radio-labelled pharmaceuticals. The image noise level and resolution of SPECT images are often poor, due to the limited number of detected counts and various physical degradation factors during SPECT acquisition (Garcia, 2012). This problem has considerably affected lesion detection, clinical diagnosis and treatment.

Recently generative adversarial networks (GAN) have been proved successfully in numerous computer vision tasks such as super-resolution, synthesis and denoising for imaging (Creswell et al., 2018), showing better performance comparing to traditional methods when applying abundant training data. Some researchers have also applied this state-of-art method in CT denoising and demonstrated ideal results without complex procedures (Wolterink et al., 2017; Yang et al., 2018). However, using GAN method for reducing noise level in SPECT images is still under explored (Mok et al., 2018; Ramon et al., 2018).

In this paper, we aim to apply and evaluate the use of GAN method in static SPECT image denoising to reduce the injection dose based on 10 simulated patient datasets.

## 2. Method

*Dataset Generation*

In order to training and testing proposed network, the 4D Extended Cardiac Torso (XCAT) phantom (Segars et al., 2010) was used to simulate 10 male and female patients with different organ sizes and activity uptakes (Figure 1).Nine phantoms were selected for training, while one phantom was chosen for testing. An analytical projector was applied to simulate 120 projections from right anterior oblique to left posterior oblique with two noise levels. The first noise level was based on a standard clinical count rate of 987 MBq injection and 16 min acquisition (low noise) while the other was 1/8 of the previous count rate (high noise). The projections were based on a low energy high resolution collimator, modelling detector-collimator response and attenuation and were then reconstructed by the ordered subset

expectation maximization (OS-EM) algorithm with 5 iterations and 6 subsets, using the cine average CT for attenuation correction. The reconstruction matrix size is $128 \times 128 \times 114$.

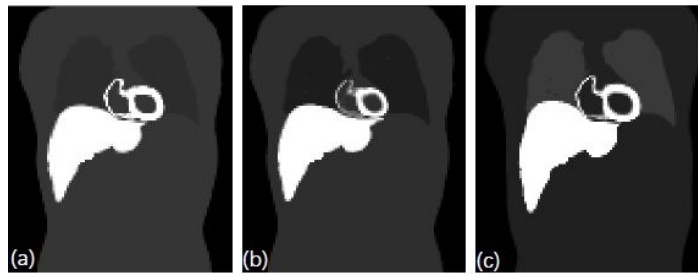

Figure 1: Three selected XCAT phantoms used in this study.

*Generative Adversarial Network (GAN)*

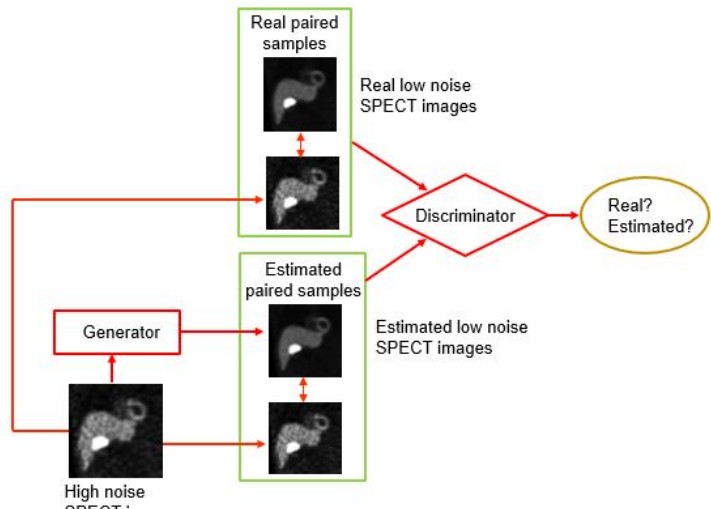

Figure 2: The conditional GAN used in this work.

Generative Adversarial Network (GAN) is a method of unsupervised learning using two neural networks against each other (Goodfellow et al., 2014). It consists of a generative network (generator) and a discriminant network (discriminator). The generator takes random sampling from latent space as input, and its output imitates the real samples in the training set. The discriminator aims to distinguish the real sample from the output of the generator. The two networks work against each other and constantly adjust their parameters. The final goal is to make the discriminator unable to discriminate the output of the generator from the real images. Conditional GAN is formed when the input of the original GAN is conditioned with additional information (Isola et al., 2017) and is used in this study (Figure 2). The high noise SPECT images were input to the generator while the discriminator compares the generated samples with the real samples, i.e., the low noise SPECT images. The calculated loss, i.e., the difference between the generated images and the real samples, would be used for tuning the generator and discriminator simultaneously. This conditional GAN was implemented in Torch and ran on a NVIDIA GeForce GTX 1070 GPU. Both generator and discriminator were optimized by using the Adam optimizer with

a learning rate of 0.00001 and 800 training epochs. The total training time was 2.7 hrs. The high noise and low noise SPECT images of nine patients, i.e., a total of 1026 images (9×114 axial slices) respectively, were paired for training (Figure 3) while 1 patient with high noise SPECT images were tested using the trained conditional GAN.

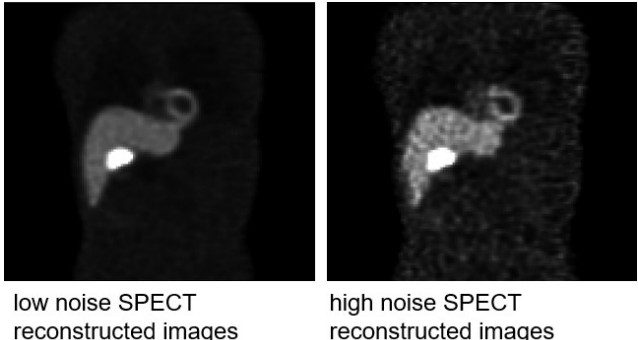

low noise SPECT
reconstructed images

high noise SPECT
reconstructed images

Figure 3: A sample pair of low noise and high noise SPECT used for training the conditional GAN.

*Data Post-processing and Analysis*

The noise level is measured by the normalized standard deviation (NSD) on a 2D uniform region-of-interest (ROI) with 82 pixels on the liver, in order to compare the results for with and without conditional GAN denoising on the tested SPECT reconstructed images.

## 3. Results and Conclusion

The noise level is substantially reduced in high noise SPECT reconstructed images after using the GAN. The NSD values are 0.1213 and 0.0693 respectively for without and with denoising (Figure 4a and Figure 4b). The NSD value of the low noise SPECT images is 0.0502 (Figure 4c).

This proposed method has the potential to decrease the noise level of SPECT images, leading to a reduced injection dose or acquisition time while still maintaining the similar image quality as compared to the original low noise images for clinical diagnosis. Further investigation of this method using the clinical SPECT patients datasets are warranted.

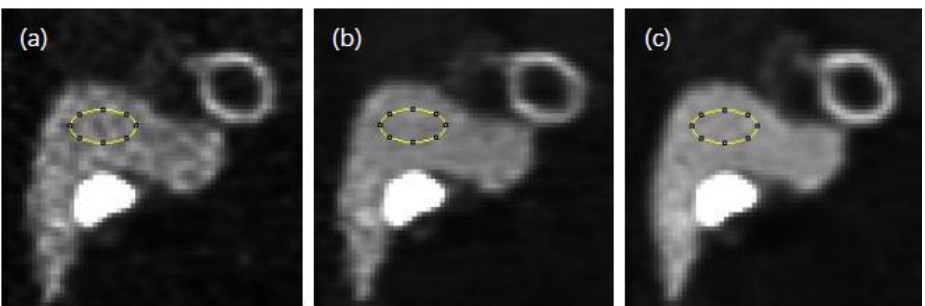

Figure 4: Sample high noise reconstructed images (a) before using GAN, (b) after using GAN and (c) the original low noise level.

## Acknowledgments

This work was supported by the research grants from the National Natural Science Foundation of China (NSFC), China (81601525), Science and Technology Fund (FDCT) of Macau (114/2016/A3) and University of Macau, Macau (MYRG2016-00091-FST).

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
