# OpenReview forum: "Low dose SPECT image denoising using a generative adversarial network"
_MIDL.io/2019/Conference/Abstract — MIDL Abstract 2019_

### Official Review · AnonReviewer1 · 2019-04-29
**An application of the conditional GAN to denoising SPECT images**

**Rating:** 3
**Confidence:** 2

**Review:**

The authors demonstrate how a conditional GAN can be used to denoise low-dose SPECT images. The evaluation on the phantom data is a good idea as this would circumvent the issue of motion-related deformation and misalignment between the target and input images, which can confound evaluation.

---

### Official Review · AnonReviewer2 · 2019-05-01
**Interesting work on a new area, but with a few minor quibbles**

**Rating:** 3
**Confidence:** 2

**Review:**

This abstract presents a novel method to denoise SPECT images using generative adversarial networks. The generator is trained to take a high-noise SPECT image and output a "cleaned" version, which might be obtained from a better acquisition protocol. The discriminator is trained to distinguish between an actual low-noise SPECT image and the generator output. The XCAT phantom is used to simulate both low- and high-noise SPECT from ten patients. The GAN is trained on nine patients and tested on the held-out case. The results show a significant decrease in normalized standard deviation after the GAN.

Overall, this is a simple but interesting idea with promising preliminary results. Therefore, I believe it would make a good addition to the MIDL conference and recommend acceptance. However, I do have some minor concerns, which the authors should address as this work matures:

1) The method, as described, is a regular GAN and not a conditional GAN. Namely, the generator learns a single data distribution (i.e., the perturbation from high- to low-noise). There are no auxiliary variables, such as age, slice, body part, etc. that could act as a conditional input.

2) How prone is this model to over-fitting? There are only 1026 image pairs to learn the entire 2D slice. An alternative would be to learn a patch-based reconstruction, which would increase the training data while lowering the parameterization.

3) Based on Figure 4 it seems that the GAN acts like a smoothing operator. Therefore, I would like to see a quantitative comparison between the outlined method and a simple 2D blurring function.

---

### Decision · Program_Chairs · 2019-05-06
**Acceptance Decision**

Accept